# Evolution of the Industrial Innovation Ecosystem of Resource-Based Cities (RBCs): A Case Study of Shanxi Province, China

**Jun Yao [1]**[ID]**, Huajing Li [1,*], Di Shang [2] and Luyang Ding [1]**

[1] School of Economics and Management, Beijing Forestry University, Beijing 100083, China; yaojun0229@163.com (J.Y.); dingluyang0212@163.com (L.D.)

[2] School of Economics and Management, Harbin Institute of Technology (Shenzhen), Shenzhen 518055, China; shangdi0215@163.com

\* Correspondence: lihuajing@bjfu.edu.cn; Tel.: +86-010-6233-8439

**Abstract:** Constructing and exploring the evolution mechanism of an industrial innovation ecosystem in resource-based cities (RBCs) is the most effective way to solve the contradiction between economic development, energy shortage, and environmental degradation. Taking 10 typical RBCs in Shanxi Province as examples, this paper used the method of system dynamics (SD) to build a model of the industrial innovation ecosystem of RBCs and set up scenarios to simulate and predict the evolution of the industrial innovation ecosystem of RBCs. The results showed that the industrial innovation ecosystem of RBCs is a complex system composed of four subsystems: innovation players, innovation content, innovation resources, and innovation environment. In innovation players, the increase in the amount of talent has a more obvious effect on technology level and GDP than R&D funding. In innovation content, the improvement of management level has a slow and continuous positive impact on GDP. Technology achievements, once implemented, will improve GDP more than management progress does. In innovation resources, human capital has greater potential for an increase in GDP and per capita consumption expenditure. In innovation resources, technology level plays an important role in slowing down the deterioration of the ecological environment. This study enriched the theoretical paradigm of the research on the industrial innovation ecosystem, and provided effective strategies to solve the development problems of RBCs.

**Keywords:** energy shortage; industrial innovation ecosystem; resource-based cities; system dynamics; evolution

## 1. Introduction

In the context of green development, innovation activities can not only promote the efficiency of resource allocation [1] and enhance the core competitiveness [2], but can also drive the high-quality socio-economic development of the country [3]. As green development has become a global consensus for development, an increasing number of countries are engaged in "carbon peak" and "carbon neutral" by adjusting their energy structure, and transforming and upgrading their industrial structure [4]. The 14th Five-Year Plan for National Economic and Social Development of the People's Republic of China and the Outline of Long-Term Goals for 2035 emphasize that it is necessary to develop and expand strategic emerging industries and to accelerate the green transformation of the development pattern [5]. However, due to technical problems, energy production and $CO_2$ emission reduction in China will face ongoing challenges [6]. Exploring the evolution of the industrial innovation ecosystem is imperative for enhancing the core industrial competitiveness and achieving sustainable socio-economic development [7].

The industrial innovation ecosystem evolved from the innovation ecosystem, and the idea was inspired by ecosystem diversity [8]. An innovation ecosystem refers to a system

where companies integrate their own products to form "customer-oriented" plans through collaborative efforts. The system is made up of customers, suppliers, and companies that co-operate with and compete against each other to secure survival and a competitive edge [9]. Since the early 1990s, based on the research on innovation ecosystems, many scholars have begun to pay attention to the innovation ecosystem from the industrial perspective, and the industrial innovation ecosystem came into being accordingly. This paper holds that driven by technological innovation elements, the industrial innovation ecosystem is a symbiotic, competitive, and dynamic evolution system shaped by the interaction between innovation players within the system. The industrial innovation ecosystem is an important foundation and key path for building and developing a new system of modern industry [10].

Resource-based cities (RBCs) are cities featuring the mining and processing of natural resources such as minerals and forests [11]. Revenues from resources in RBCs often fail to foster broader socio-economic development. On the contrary, they have been shown to adversely affect some countries' economic, social, and environmental well-being [12], which is the "resource curse" phenomenon [13,14]. RBCs in various countries adopt different strategies to deal with the challenge of the resource curse. For many Indigenous Australians, they entail full forest resource utilization ahead of mining and multiple-use mine rehabilitation for long-term environmental, cultural, and livelihood benefits [15]. In Mexico, to avoid threatening the environment and health of people caused by mining waste, the remote-sensing technique was used to identify waste [16]. The orderly development of urban Enugu (a coal city in Nigeria) demands upgrading the slums and squatter settlements [17]. In Brazil, financial mechanisms were used to compensate for the negative environmental impacts of mining. There are 262 RBCs in China [18]. RBCs, as strategic energy resource bases, are important supports for the sustained and sound development of the national economy [19]. Political choices and regulation can help maintain high levels of social, cultural, and environmental values for RBCs [20,21]. Industrial structure [22], regional distribution [23], degree of opening to the outside world [24], technology innovation [25], and other aspects also affect the sustainable development of RBCs.

As a typical resource-based region in China, Shanxi Province is an important energy supply base [26] (Figure 1). Except for Taiyuan, the capital city of Shanxi Province, there are 10 administrative districts on the list of national RBCs, including Datong, Shuozhou, Yangquan, Changzhi, Jincheng Xinzhou, Jinzhong, Linfen, Yuncheng, and Lvliang. The long-standing extensive RBC development model leads to irreparable modern city problems such as the worsening shortage of energy resources and the destruction of the ecological environment [27]. Therefore, constructing the evolution model of the industrial innovation ecosystem of RBCs and exploring its evolution mechanism is of great significance to solve the contradictions of industrial and economic development with resources and the environment.

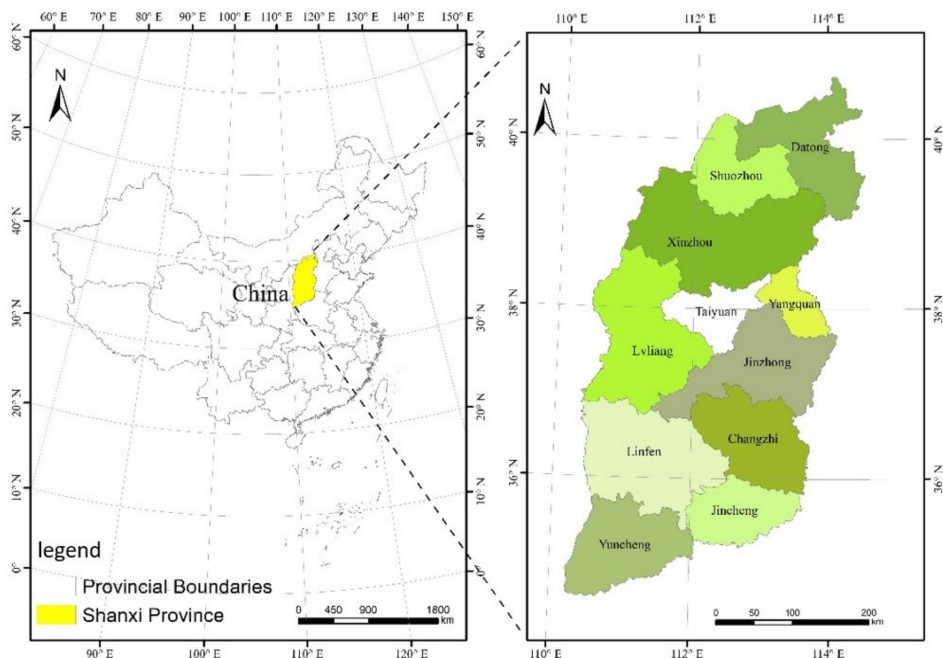

**Figure 1.** Geographic locations of the investigated areas (10 RBCs in Shanxi Province).

## 2. Materials and Methods

### 2.1. Research Method

System dynamics (SD) is a science that studies the structure and behavior of system feedback, which integrates system science with computer simulation. SD was first proposed by Jay W. Forrester, a professor at Massachusetts Institute of Technology in the United States in the 1950s. In the early stage, SD was mainly applied to the management of industrial companies. In the 1990s, SD was wildly spread to major socio-economic fields such as macro-economics, project management, and corporate strategy. SD can scientifically simulate and predict the dynamic evolution of complex systems by means of standardized computer simulation systems, which can help scholars better understand the structure and dynamic behavior of complex systems. The dynamic evolution of the complex industrial innovation ecosystem is affected by the components and external environment of the system. Therefore, it is scientific and feasible to analyze the dynamic evolution of the industrial innovation ecosystem by using an SD model.

### 2.2. Data Sources

Based on data from 2010 to 2019 on RBCs in Shanxi Province, this paper collected and sorted out relevant indices, conducted mathematical modeling through Vensim, and made empirical analyses. To ensure the accuracy and availability of the data, the model parameters were mainly constructed on the basis of the historical data from 2010 to 2019.The data were selected from official documents like the China Statistical Yearbook, the Shanxi Statistical Yearbook, the China Urban Statistical Yearbook, the China Statistical Yearbook on Science and Technology, and the Shanxi National Economic and Social Development Statistical Bulletin. Given that SD modeling emphasizes the composition of the model rather than the accuracy of the parameters, some quantified variables were obtained through reasonable estimation and correction.

### 2.3. SD Modeling

#### 2.3.1. Construction of the Causality Model

Based on analyses of the overall structure, internal factors, and external environment of the industrial innovation ecosystem of RBCs, this paper explored the extensive evolution of the complex industrial innovation ecosystem of RBCs from the perspective of causality

of four subsystems, i.e., innovation players, innovation content, innovation resources, and innovation environment.

### 2.3.2. Causality Analysis of Innovation Players

Innovation players consist of the government, companies, universities and research institutions, and intermediary service agencies. The government provides favorable economic and regulatory conditions for companies, universities, and research institutions [28]. The government can not only offer hard security by improving infrastructure such as transportation and communications, but also provide soft support by establishing and improving relevant laws and regulations, such as tax reduction and exemption policies [25]. Companies are the core players. They transform human capital, technology innovation, and other elements of production into innovative products and launch these products to the target market to obtain economic benefits. The universities and research institutions, as fundamental sources of knowledge, can contribute to open innovation dynamics in the ecosystem through granting licensing technology to the industry and promoting the creation of spin-offs [29]. Acting as a bridge, intermediary service agencies are responsible for collecting innovative information through multiple channels. They are supposed to facilitate the diffusion of innovative information and promote the flow of innovative elements.

### 2.3.3. Causality Analysis of Innovation Content

Innovation involves technology innovation and management innovation. Technology innovation is the endogenous driving force for the evolution of the industrial innovation ecosystem of RBCs [30,31]. It can transform knowledge capital into productivity and thereby promote corporate to reform through technology diffusion. Combining technological innovation with management innovation can jointly improve the industrialization level of enterprises. Management innovation refers to the introduction of new management elements into the corporate management system, thus achieving organizational goals and innovation activities more effectively. Companies in RBCs are mainly mineral and forest mining companies [32]. The old-fashioned management concept of "preferring technology to management" has led to the long-term neglect of management innovation. Therefore, companies in RBCs should learn from or develop new management experience or ideas, and reinforce their ability to engage in independent innovation.

### 2.3.4. Causality Analysis of Innovation Resources

Innovation resources cover funds, human capital, and information. The entire innovation process, ranging from innovation project launching and innovation production, to innovation achievement commercialization, is inseparable from the support of funds [33]. Human capital is the core of the industrial innovation ecosystem of RBCs. Driven by innovation mentality, talents can fully integrate innovation elements and information resources, and inspire the innovation players to develop new technologies. Information is the carrier of innovation activities. Knowledge diffuses to all phases of the innovation process through flows of information, including phases of R&D, application, and implementation [34].

### 2.3.5. Causality Analysis of Innovation Environment

The innovation environment is composed of social, cultural, and ecological environments. From the perspective of society and culture, positive innovation culture can attract the accumulation of human capital, material capital, and other innovation elements, thus improving the supply quality and efficiency of innovation resources. The ecological environment of RBCs plays an irreplaceable role in the formation of their industrial innovation ecosystem. For one thing, the economic development of RBCs is subject to the natural environment, and as a result, their innovation products are resource-specific. For another thing, the ecological environments of some RBCs have been damaged due to long-term extensive mining, so it is necessary to develop an environmental governance mode by

technology innovation [35]. From the macro coevolution view, an ecosystem permanently exchanges with environments for continuous innovation [36,37].

Based on the above analyses of the evolution of the industrial innovation ecosystem of RBCs from four subsystems, this paper set GDP, technological level, and total labor force as state variables, and set government R&D investment, R&D investment from universities and research institutions, investment in mining environment restoration, emission of main pollutants, annual resource output, and management level as auxiliary variables. On this basis, this paper drew a dynamic flow chart of the industrial innovation ecosystem development of RBCs (Figure 2). The system dynamics flow diagram of the industrial innovation ecosystem of RBCs includes four subsystems: innovation players, innovation content, innovation resources, and innovation environment. The feedback loop of each subsystem includes:

(1) Innovation players subsystem: R&D investment/talent investment → technical level → GDP → general budget of financial revenue → R&D investment/talent investment.

(2) Innovation content subsystem: technical level → GDP → general budget of financial revenue → R&D investment/talent investment → technical level, management level → GDP → R&D investment/talent investment → total enterprise profit→management level.

(3) Innovation resource subsystem: primary energy production → GDP → technical level→primary energy production,; total labor force → GDP → technical level → reduction rate of death in production safety accidents/number of shanty towns completed → total labor force → GDP, GDP → per capita GDP → per capita disposable income → per capita consumption expenditure → GDP.

(4) Innovation environment subsystem forms negative feedback mechanisms such as primary energy production → emission of major pollutants → environmental quality and number of enterprises → environmental quality, and positive feedback loops such as GDP → general budget of financial revenue → technical level → expenditure for energy conservation and environment restoration → environmental quality → total labor force → GDP.

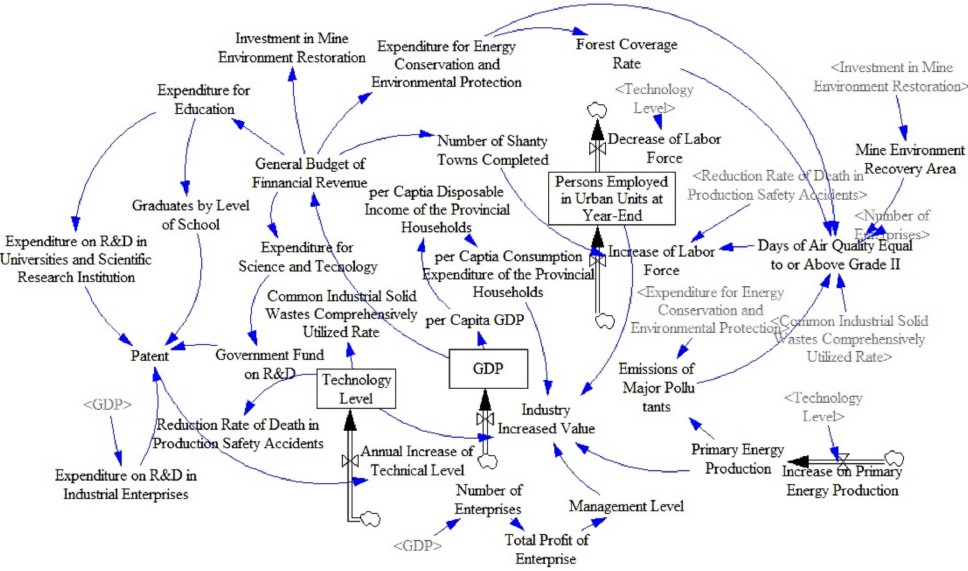

**Figure 2.** SD flow chart of the industrial innovation ecosystem development of RBCs.

*2.4. Simulation Analysis*

2.4.1. Model Validation

The simulation analysis of this study was run by Vensim. It was assumed that the running time was from 2010 to 2019, and the simulation step was one year. To test whether the operation of the model conforms to the reality, this paper took the GDP of the 10 RBCs

in Shanxi Province as an example, and compared the simulation data of the model with the existing historical data. According to the production law, GDP is equal to the sum of the added value of the primary, secondary, and tertiary industries. As shown in the Figure 3, in the industrial innovation ecosystem in RBCs, the change trend of GDP simulation value was close to that of the real value, and its goodness of fit was over 88%. Therefore, the industrial innovation ecosystem model of RBCs can objectively reflect the industrial and socio-economic development of RBCs in Shanxi Province from 2010 to 2019, which indicates that the model is sound and precise. So, the model was employed to simulate and predict the formation and evolution of the industrial innovation ecosystem of RBCs in Shanxi Province.

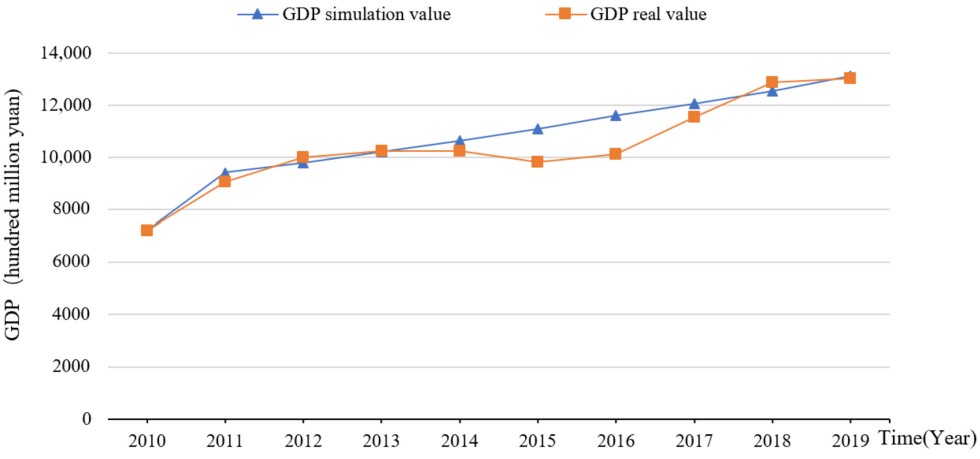

**Figure 3.** Comparison of simulated GDP and real GDP.

### 2.4.2. Simulation

To analyze the evolution of the industrial innovation ecosystem of RBCs, the SD model was adopted to build the model of industrial innovation ecosystem of RBCs. This paper explored the mechanism and the effects of changes in key element investment on GDP, per capita consumption expenditure, and environmental quality. In this paper, it was found that the initial value of the ideal coefficient of government R&D investment was close to 1.5 in the repeated modification of the model. The initial value of other variable coefficients was 1. The simulation situation was set by adjusting the coefficients of different variables in the four subsystems of innovation subject, innovation content, innovation resources, and innovation environment. The focus of the paper was to explore the effect of increasing the input of different variables on the industrial innovation ecosystem of RBCs. Therefore, referring to the setting of situational mode in the existing literature [38,39], this paper increased it by 0.5 times on the basis of the initial mode coefficient when setting the situational mode of subsystem.

### 3. Results

#### 3.1. Simulation of Innovation Players

For innovation players, the coefficients of government R&D investment, university and research institution R&D investment, corporate R&D investment, and amount of talent in universities and research institutions were set as policy variables. The values of the four policy variables were adjusted to form different situational modes (Table 1), and the operation of the model under different situational modes was observed. In Situational Mode 1-1, the coefficient of government R&D investment increased, whereas the coefficients of other policy variables remain unchanged. Such a mode was to observe the impact of improving government R&D investment on the technology level and GDP of RBCs. Similarly, in Situational Modes 1-2, 1-3, and 1-4, the coefficients of R&D investment from universities and research institutions, corporate R&D investment, and amount of talent

in universities and research institutions increased, whereas the rest remained unchanged. Such a mode aimed to measure the impact of the three variables on the technology level and GDP of RBCs. Situational Mode 1-5 was set to observe the impact on the technology level and GDP of RBCs by improving the four variables mentioned above simultaneously. The simulation results of technology level and GDP in the five situational modes regarding innovation players is shown in Figures 4 and 5, respectively.

In terms of innovation players, as shown in Figures 4 and 5, increasing government R&D investment, R&D investment from universities and research institutions, corporate R&D investment, and the amount of talent in universities and research institutions, individually or simultaneously, had a significantly positive effect on the improvement in technology level and industrial added value to different degrees. The orders from highest to lowest degrees were increasing players simultaneously (Situation Mode 1-5), only increasing the amount of talent in universities and research institutions (Situation Mode 1-4), only increasing corporate R&D investment (Situation Mode 1-3), only increasing government R&D investment (Situation Mode 1-1), and only increasing R&D investment from universities and research institutions (Situation Mode 1-2).

**Table 1.** Variable setting in the situational modes regarding innovation players.

|  | Government R&D Investment | R&D Investment from Universities and Research Institutions | Corporate R&D Investment | Amount of Talent in Universities and Research Institutions |
|---|---|---|---|---|
| Initial Mode 1-0 | 1.5 | 1 | 1 | 1 |
| Situational Mode 1-1 | 2 | 1 | 1 | 1 |
| Situational Mode 1-2 | 1.5 | 1.5 | 1 | 1 |
| Situational Mode 1-3 | 1.5 | 1 | 1.5 | 1 |
| Situational Mode 1-4 | 1.5 | 1 | 1 | 1.5 |
| Situational Mode 1-5 | 2 | 1.5 | 1.5 | 1.5 |

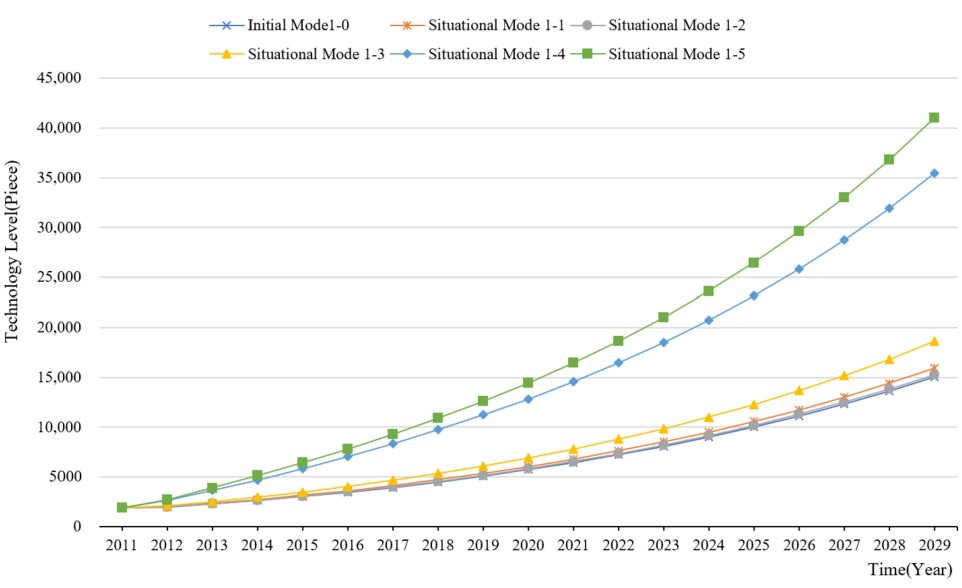

**Figure 4.** Simulation results of technology level in situational modes regarding innovation players.

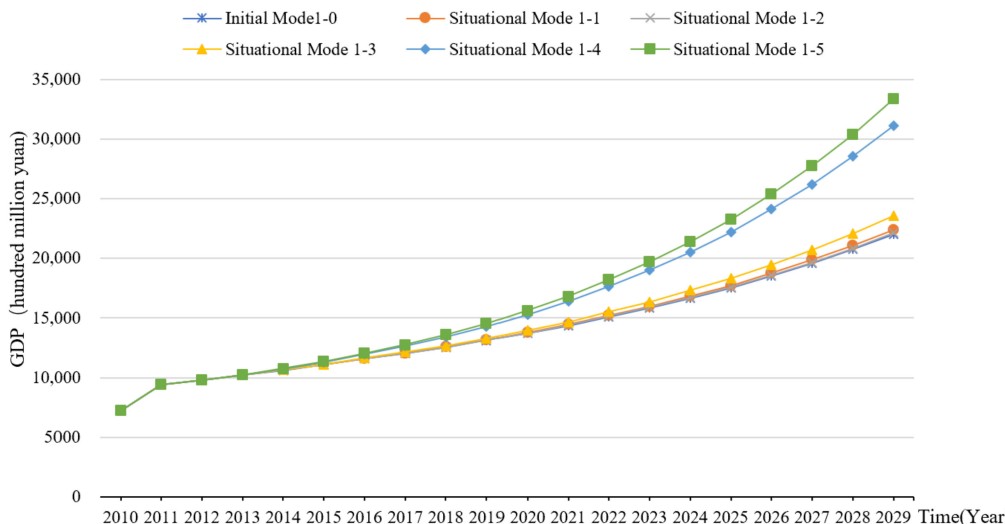

**Figure 5.** Simulation results of GDP in situational modes regarding innovation players.

### 3.2. Simulation of Innovation Content

In terms of innovation content, this paper took the coefficients of investment in enterprise management level and enterprise technology level as policy variables, adjusted them to form different situational modes (Table 2), and observed the operation of the model in different situational modes. In Situational Mode 2-1, the coefficient of investment in enterprise management level increased, whereas investment in enterprise technology level remained unchanged. Such a mode aimed to observe the impact of improving enterprise management level on the GDP of RBCs. In Situational Model 2-2, the coefficient of enterprise technology level increased by the approach stated in Situational Mode 1-6, but the coefficient of investment in enterprise management level remained unchanged. Such a mode aimed to observe the impact of improving enterprise technology level on the GDP of RBCs. In Situational Mode 2-3, the investment coefficients in both enterprise management level and enterprise technology level increased at the same time, which aimed to observe the impact of improving the two aspects on RBCs' GDP. The simulation results in these three situational modes regarding innovation content can be seen in Figure 5.

**Table 2.** Variable setting in different situational modes regarding innovation content.

|  | Investment in Enterprise Management Innovation Level | Investment in Enterprise Technology Level |
| --- | --- | --- |
| Initial Mode 2-0 | 1 | 1 |
| Situational Mode 2-1 | 1.5 | 1 |
| Situational Mode 2-2 | 1 | 1.5 |
| Situational Mode 2-3 | 1.5 | 1.5 |

Regarding innovation content, as shown in Figure 6, increasing the investment coefficients in both enterprise management innovation level and enterprise technology level had significant positive effects on GDP growth, but the effects on GDP were different. The crossover would take place in 2023, before which the effect of management level would be greater than that of technology level, but after which the case would be the opposite. The simultaneous improvement of management level and technology level (Situational Mode 2-3) would most significantly increase GDP.

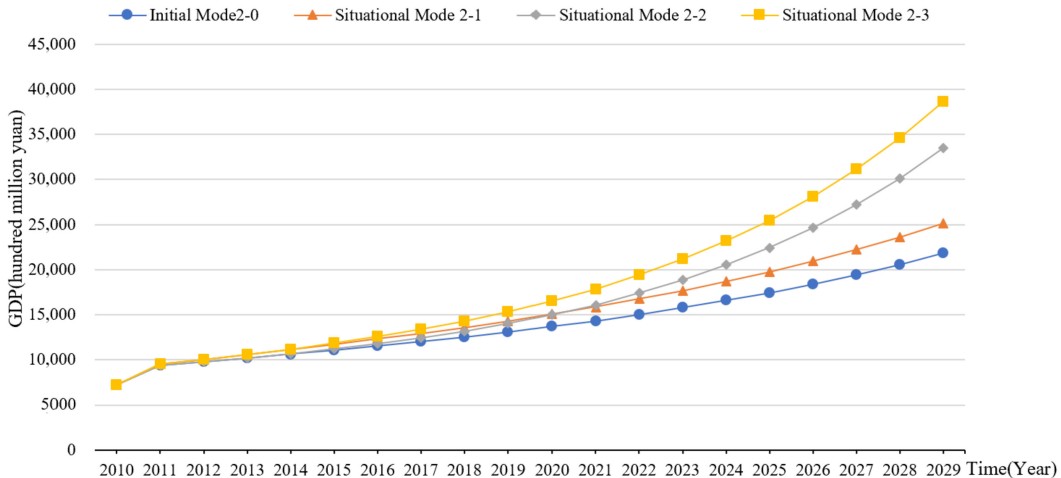

**Figure 6.** Simulation results of GDP in different situational modes regarding innovation content.

### 3.3. Simulation of Innovation Resources

In terms of innovation resources, since it is difficult to quantify capital resources and information resources, only human capital (total labor force) was selected for quantitative research (Table 3). Considering that environmental quality and the reduction rate of production safety accidents have little impact on total labor force, only the number of completed shantytowns was selected to study total labor force (Situational Mode 3-1). The impact of the variation of total labor force on the GDP and per capita consumption expenditure of RBCs was also worth considering (Situational Mode 3-2).

Regarding innovation resources, as shown in Figure 7, total labor force increased with the number of completed shantytowns. However, in the long run, the increase in labor force showed a relatively slow upward trend. As shown in Figure 8, increasing the investment in human capital can contribute to GDP growth. Figure 9 indicates that increasing human capital investment also had a significantly positive impact on the increase in per capita consumption expenditure.

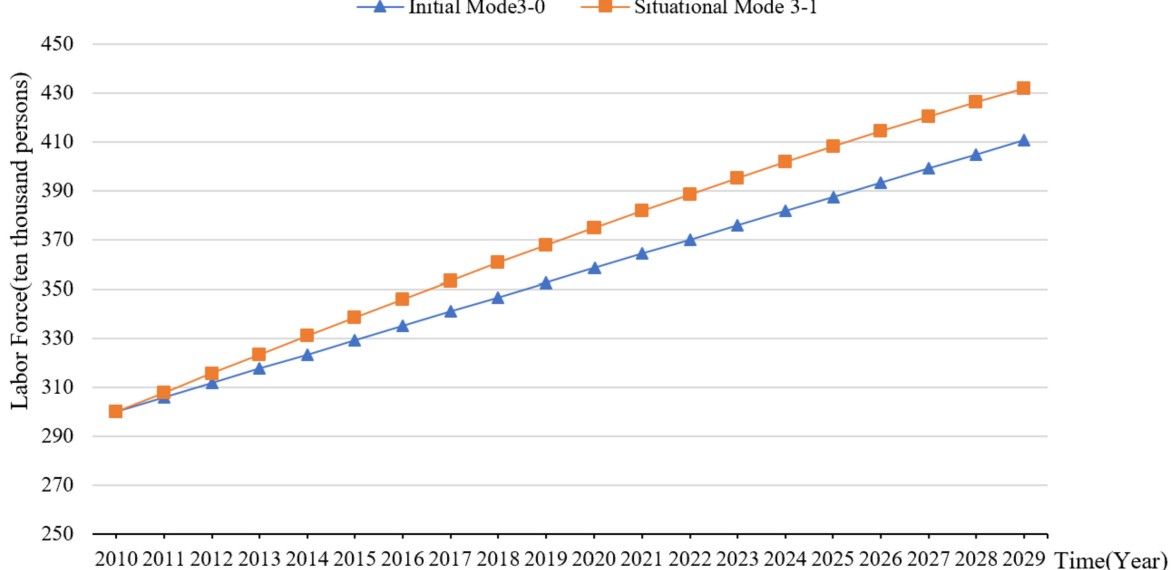

**Figure 7.** Simulation results of labor force in different situational modes regarding innovation resources.

**Table 3.** Variable setting in different situational modes regarding innovation resources.

| | Number of Completed Shantytowns | Total Labor Force |
|---|---|---|
| Initial Mode 3-0 | 1 | 1 |
| Situational Mode 3-1 | 1.5 | 1 |
| Situational Mode 3-2 | 1 | 1.5 |

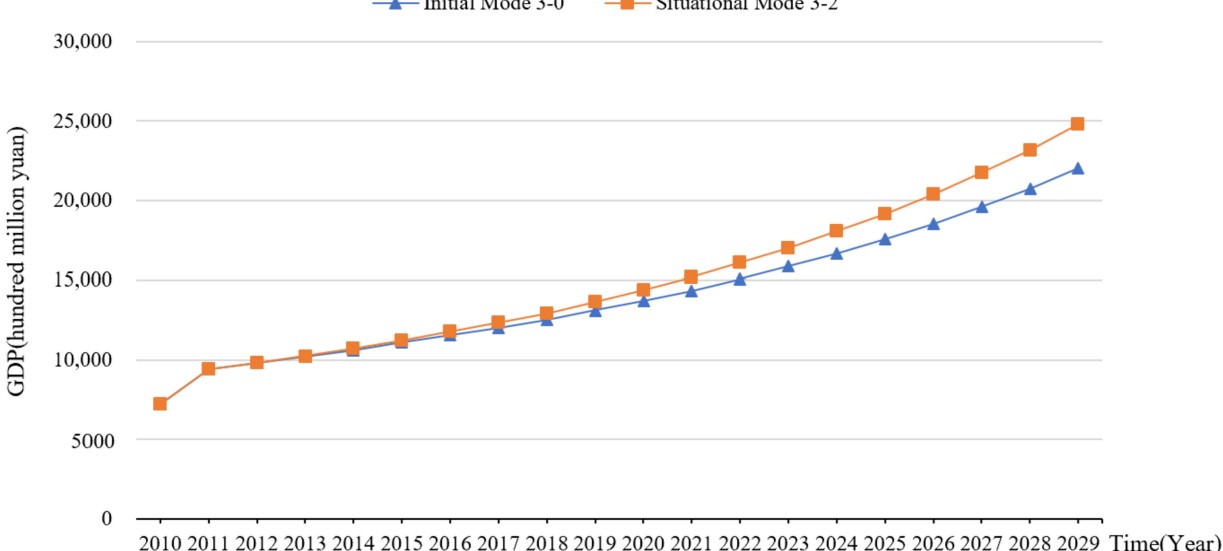

**Figure 8.** Simulation results of GDP in different situational modes regarding innovation resources.

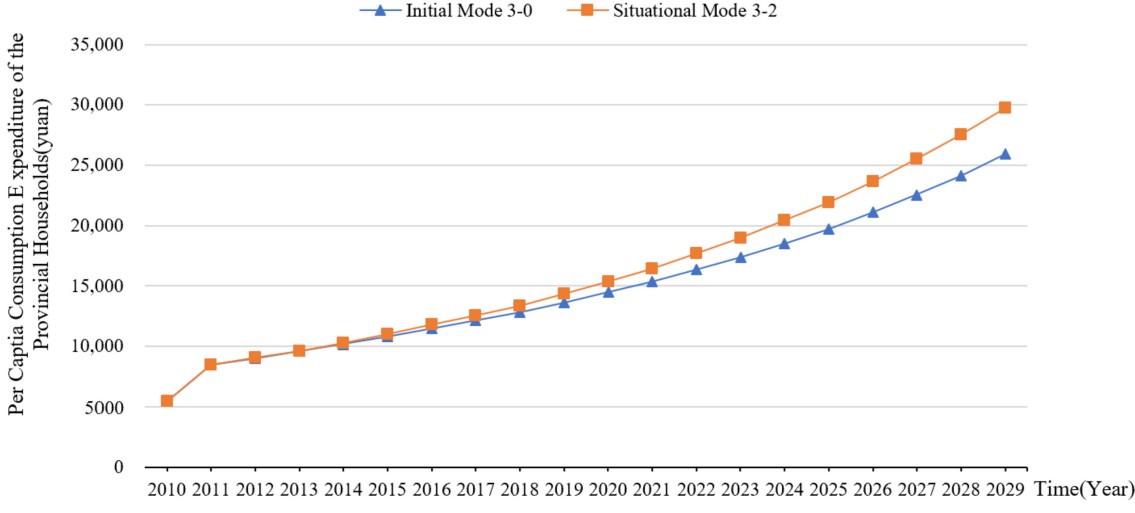

**Figure 9.** Simulation results of per capita consumption expenditure of provincial households in different situational modes regarding innovation resources.

### 3.4. Simulation of Innovation Environment

In terms of innovation environment, the investment in environmental management, annual resource output, technology level, investment in environment restoration, and comprehensive utilization rate of industrial solid waste were set as variables, and the values of the above five variables were adjusted to form different situational modes (Table 4). The number of days of air quality reaching the standard was used to measure environmental quality. The operation of the model was observed in different situational modes. Situational Mode 4-1 was to observe the impact of the increase in investment in environmental manage-

ment on the variation of forest coverage. Situational Mode 4-2 was to observe the impact of the increase in annual resource output on environmental quality. Situational Mode 4-3 was to improve the technical level based on the increase in annual resource output and observe the impact on environmental quality. Situational Mode 4-4 was to observe the impact of the increase in investment in mining environment restoration on environmental management. Situational Mode 4-5 was to observe the impact of the increase in the comprehensive utilization rate of industrial solid waste on environmental management.

**Table 4.** Variable setting in different situational modes regarding innovation environment.

| | Investment in Environmental Management | Annual Resource Output | Technology Level | Investment in Mining Environmental Restoration and Governance | Comprehensive Utilization Rate of Industrial Solid Waste |
|---|---|---|---|---|---|
| Initial Mode 4-0 | 1 | 1 | 1 | 1 | 1 |
| Situational Mode 4-1 | 1.5 | 1 | 1 | 1 | 1 |
| Situational Mode 4-2 | 1 | 1.5 | 1 | 1 | 1 |
| Situational Mode 4-3 | 1 | 1.5 | 1.5 | 1 | 1 |
| Situational Mode 4-4 | 1 | 1 | 1 | 1.5 | 1 |
| Situational Mode 4-5 | 1 | 1 | 1 | 1 | 1.5 |

Regarding innovation environment, as shown in Figure 10, increasing investment in environmental management significantly increased the forest coverage of RBCs. As shown in Figure 11, with the increase in the annual resource output, environmental quality was on the decline, but when adding technological factors when exploiting resources, environmental quality showed an upward trend. Figures 12 and 13 both show that increasing the investment in mining environmental restoration and improving the comprehensive utilization rate of industrial solid waste had a significant positive impact on environmental quality.

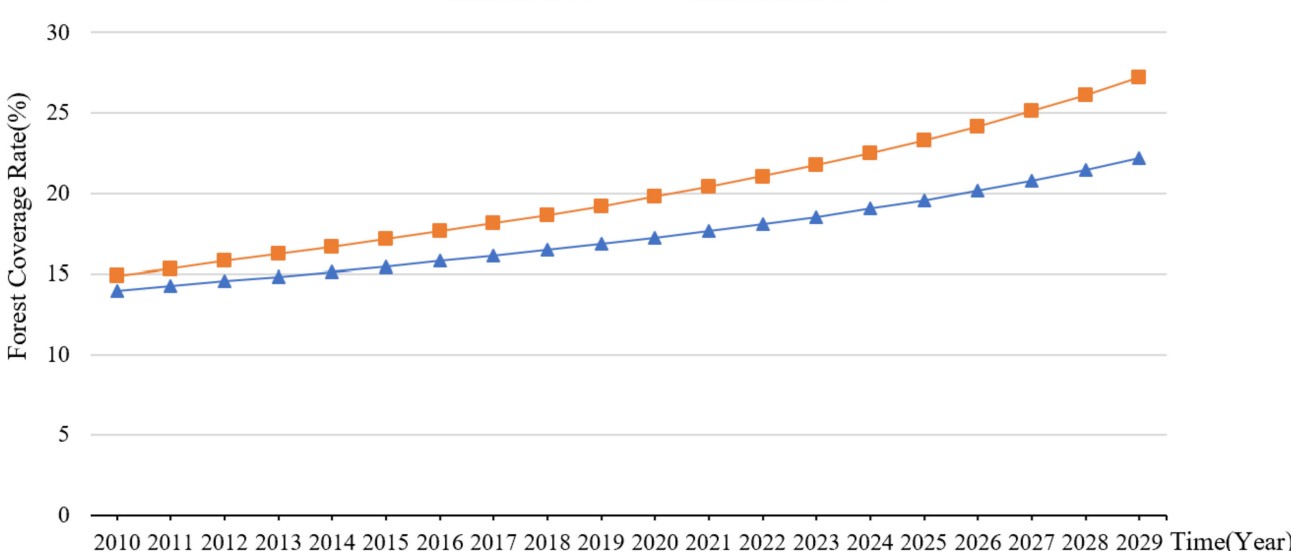

**Figure 10.** Simulation results of forest coverage rate in different situational modes regarding innovation environment.

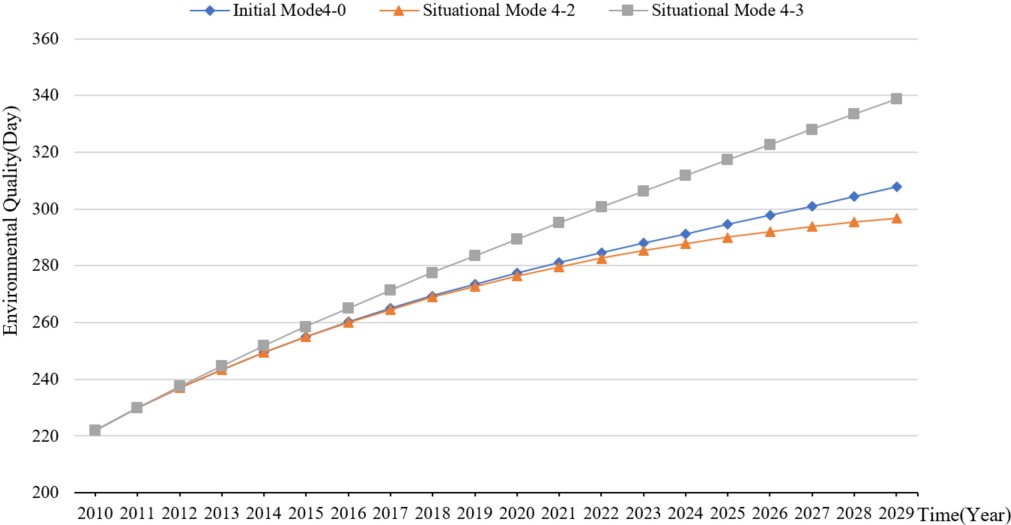

**Figure 11.** Simulation results of "Annual Resource Output-Technology Level-Environmental Quality" in dif-ferent situational modes regarding innovation environment.

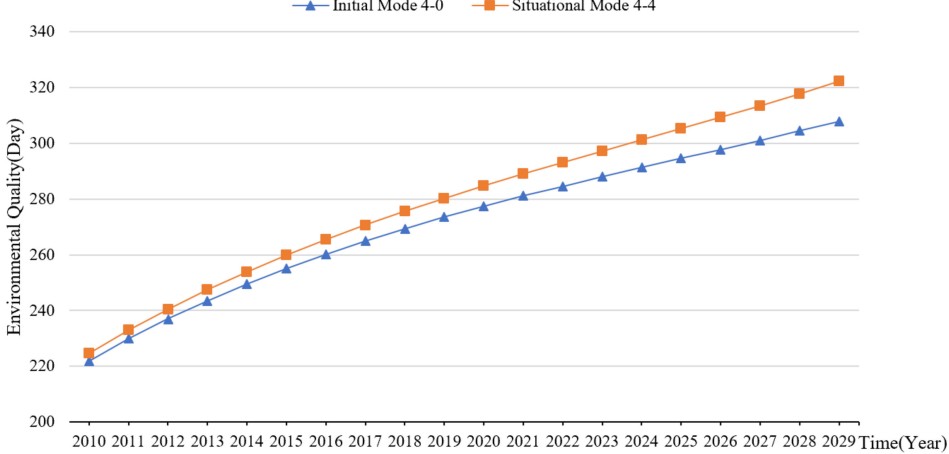

**Figure 12.** Simulation results of "Investment in Mining Environmental Restoration and Governance-Environmental Quality" in different situational modes regarding innovation environment.

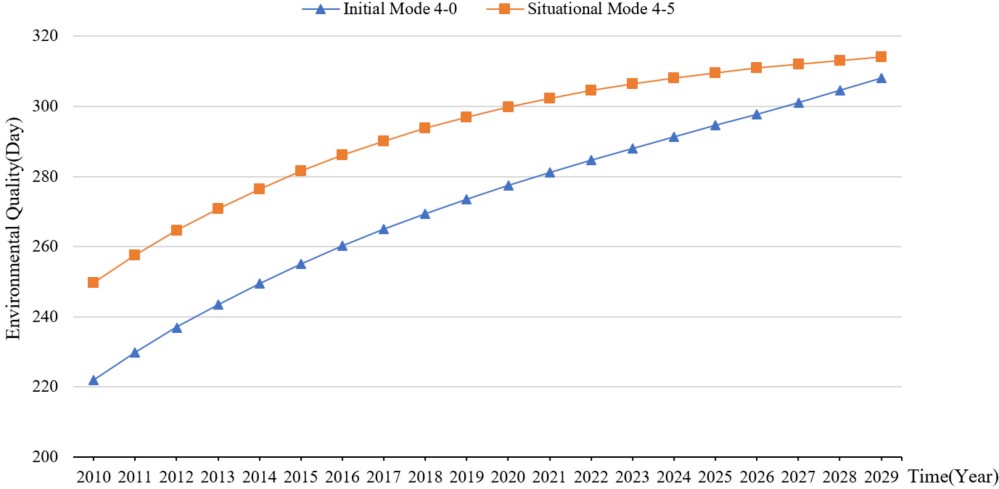

**Figure 13.** Simulation results of "Comprehensive Utilization Rate of Industrial Solid Waste-Environmental Quality" in different situational modes regarding innovation environment.

## 4. Discussion

The sound evolution of the industrial innovation ecosystem of RBCs is conducive to forming a new pattern of digital networks of RBCs and realizing sustainable and healthy development of RBCs. Meanwhile, RBCs can promote the development of surrounding areas through their own exemplary practice by providing model schemes and forming global industrial innovation clusters. Based on the above research results, this article believes the following.

First, in terms of innovation players, the increases in R&D investment from different players and amounts of talent in universities and research institutions will have a significantly positive impact on the improvement of technology level and GDP to various degrees. Human capital helps achieve innovative results through creative labor corporate R&D investment, which has the most direct effect on technology level and GDP. As the backbone of innovation activities in RBCs, companies play a decisive role in the evolution of ecological benefits of industrial innovation in RBCs. Increasing governmental R&D investment has a more significant effect than increasing R&D investment from universities and research institutions does. The government can steer policy direction, macro-regulation, and innovation.

Second, in terms of innovation content, the improvement of technology level and management level will have different positive impacts on the improvement of GDP. As management innovation targets business mode, which is directly related to management activities, the improvement has a direct, slow, and continuous positive impact on the improvement of corporate operation efficiency. However, as the impact of technology level on commercialization is often delayed, the effect may not be obvious in the early stage. Technology achievements, once implemented, will improve business operation efficiency better than management progress does.

Third, in terms of innovation resources, rural residents can go to cities through the transformation of shantytowns and thus increase urban labor force. However, in the long run, the increase in labor force is showing a relatively slow upward trend, which is attributed to the increasing aging population and the declining fertility rate in China. The expansion of higher education also delays the employment of the working-age population. Although labor force growth slows down, the improvement of education level will improve labor quality. Human capital has greater potential to improve GDP, and it also has a significant positive impact on the increase in residents' per capita consumption expenditure.

Fourth, in terms of innovation environment, the increase in investment in environmental management will lead to an increase in forest coverage in RBCs. The improvement of the comprehensive utilization rate of industrial solid waste has a significant positive impact on the environmental quality of RBCs. The increase in the annual resource output has lowered the environmental quality of RBCs. However, in resource exploitation, when increasing the investment in technology level, environmental quality shows an upward trend, indicating that in energy development and resource exploitation, technology level plays an important role in slowing down the deterioration of the ecological environment.

## 5. Suggestions

Taking the 10 RBCs in Shanxi Province as research objects, this study used a system dynamics method, selected data from 2010 to 2019, and forecasted the dynamic evolution trend of RBCs in the next 10 years from aspects of innovation players, innovation content, innovation resources, and innovation environment. The results showed that the industrial innovation ecosystem of RBCs is a complex system composed of four subsystems: innovation players, innovation content, innovation resources, and innovation environment. In innovation players, the increase in the number of talents has a more obvious effect on technology level and GDP than R&D funding. In innovation content, the improvement in management level has a slow and continuous positive impact on GDP. Technology achievements, once implemented, will improve GDP better than management progress does. In innovation resources, human capital has greater potential for an increase in GDP and

per capita consumption expenditure. In innovation resources, technology level plays an important role in slowing down the deterioration of the ecological environment. According to the conclusions of this paper, in order to realize the sustainable evolution of industrial innovation ecosystem in RBCs, this paper puts forward the following suggestions.

(a) In terms of innovation players, it is necessary to fully stimulate the innovation capacity of innovation players, and actively guide the symbiotic evolution of innovation players. First, the government should play a leading and coordinating role to enhance the external driving force for the evolution of the industrial innovation ecosystem in RBCs. Government support and policy inclination are important guarantees for evolution and development. The government should formulate a long-term development strategy and planning layout of regional industries, such as encouraging and guiding enterprises, universities, and scientific research institutions to cooperate; formulate policies on innovation protection, patent protection, tax preference, and financial support; and introduce and cultivate innovative talent. Second, it is necessary to cultivate and strengthen enterprises so as to enhance the endogenous driver of the formation and development of the industrial innovation ecosystem in RBCs. Traditional extensive enterprises should be guided to transform from high pollution, high emissions, and high energy consumption into high-tech. Platforms such as incubation bases can be established to play a leading and exemplary role in improving the environment, developing the economy, and serving society. Meanwhile, innovation elements and resources of universities, scientific research institutions, intermediaries, and service institutions should be gathered to form a mutually beneficial symbiosis model in RBCs.

(b) In terms of innovation content, it is necessary to play the leading role of technological innovation and digital technology, and further tap the strategic value of management innovation and organizational change. First, advanced management concepts and technologies should be introduced or innovated to enhance independent innovation capacity and to promote their efficiency of technological change. Second, advanced technologies should be introduced and invented to achieve resource conservation and comprehensive utilization. Resource-mining enterprises should be guided to introduce and invent to effectively improve the utilization and recycling efficiency of mineral resources. Forest industry enterprises should improve the efficiency of forest harvesting and the comprehensive utilization of processing residues, lumbering, and waste wood materials to achieve the value added.

(c) In terms of innovation resources, basic resources such as forest and mineral resources should be fully developed. Technological transformation projects should be implemented so as to realize the green transformation and upgrading of traditional industries. Firstly, big-data platforms and intelligent sensing technologies should be fully utilized to promote the intelligent construction and transformation of coal mines. Dangerous operations should be done by robots rather than people, thus improving the safety and efficiency of coal mining. Secondly, the rich natural, ecological resources and profound historical and cultural heritage of Shanxi Province should be fully utilized. Health care towns and communities should be built in Datong, Xinzhou, Jinzhong, and other cities of the province. Demonstration areas of the "health care industry" should be built to attract multi-level and cross-age consumer groups from regions around Shanxi such as Beijing, Tianjin, Hebei, and Henan. Thirdly, while the steel and coal production capacity is reduced, it is necessary to form industrial clusters including the modern coal chemical industry, semiconductor optoelectronics, and photovoltaic manufacturing by means of technological transformation. It is necessary to promote traditional industries to move forward towards high-end modern industries and to enhance the competitiveness of high-tech industries. Finally, the high-quality development of hydrogen energy and other clean energy industries should be furthered by promoting technological innovation and institutional innovation, thus forming a demonstration area for industrial transformation and upgrading.

Industry clusters including new energy vehicles, pharmaceuticals and healthcare, and solid waste utilization should be cultivated to stimulate new momentum for industrial development.

(d) In terms of the innovation environment, the requirements of ecological priority and green development must be implemented in industrial upgrading. The green development mode and green lifestyle should be promoted, thus forming and achieving the transformation from the dilemma between economic development and environmental protection to the win–win mode of coordinated development. First of all, the supply side structural reform should be persistently promoted to meet the actual and potential needs. The quality of supply must be improved, thus promoting positive incentives and survival of the fittest. The technological innovation and the scale effect should be utilized to cultivate and develop new industrial clusters. Secondly, efforts should be made to fight against pollution by promoting the green upgrading and transformation of industries, lowering energy and material consumption and reducing pollutant emissions. It is necessary to establish the new environmental governance modes represented by environmental protection industrial parks and green low-carbon intelligent innovation space. Industry cluster belts focusing on the transformation of green technology achievements should be created. Policies need to be introduced and implemented for environmental protection enterprises in terms of green technology R&D, talent introduction, and financial support. Thirdly, the development of low-carbon, clean, circular, and efficient green industries should be promoted so as to build a modern green development mode with harmonious coexistence between man and nature in RBCs.

**Author Contributions:** Conceptualization, H.L. and J.Y.; methodology, D.S.; software, L.D.; validation, J.Y. and D.S.; formal analysis, J.Y.; investigation, D.S.; data curation, J.Y.; writing—original draft preparation, J.Y.; writing—review and editing, H.L.; visualization, J.Y.; supervision, H.L.; project administration, H.L.; funding acquisition, H.L. All authors have read and agreed to the published version of the manuscript.

**Funding:** This work was supported by the National Natural Science Foundation of China (Grant Nos. 71972014, 71572016).

**Institutional Review Board Statement:** Not applicable.

**Informed Consent Statement:** Not applicable.

**Data Availability Statement:** Data were selected from "China Statistical Yearbook" (2010–2019), "Shanxi Statistical Yearbook" (2010–2019), "China Urban Statistical Yearbook" (2010–2019) and "China Statistical Yearbook on Science and Technology" (2010-2019), at https://www.cnki.net/ accessed on 16 April 2021; "Shanxi National Economic and Social Development Statistical Bulletin" at http://www.shanxi.gov.cn/ accessed on 16 April 2021.

**Conflicts of Interest:** The authors declare no conflict of interest.

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
