# Peer review of "Evolution of the Industrial Innovation Ecosystem of Resource-Based Cities (RBCs): A Case Study of Shanxi Province, China"

_sustainability, doi:10.3390/su132011350_

Round 1

Reviewer 1 Report

The article entitles “Evolution of the industrial innovation ecosystem of resource-based cities (RBCs)” presents a prediction of the evolution of the industrial innovation ecosystem of resource-based cities This seems to be an exciting topic; however, the writing lacks a proper presentation. Many parts of the article need to be rewritten for appropriate understanding.

  • Read the paper carefully; there are some sentences to improve. Also, Break the lines into simple statements. (31-37).
  • The abstract is not well written. From the abstract, we cannot get the main idea and novelty of this paper. Rewrite the abstract and mention the Research method and results in the abstract and conclusion.
  • Please explain your data, such as columns of data, and their features also mention the percentage of training and testing data.
  • The theoretical framework needs to be reinforced, for example, by including more references. The system is not well-defined and vague, and the experimental results are not convincing.
  • Rename current “Conclusions and suggestions” sections as “Suggestions”. And Rewrite the Conclusion in simple and concise words.
  • Add related work section and cite the related articles about RBCs other than China as well.

Reviewer 2 Report

The topic is actual and suitable for this journal. The authors describe the System dynamics (SD) method a system theory method, widely used in practice since 1970.  The strong point is the use if real data . The weak points are very lengthy descriptions with many unnecessary aspects , as the communist plans for development, descriptions of many secondary facts and data , that are not used in the study- just used to fictively improve the value of the paper.

The paper must be shortened, the quality of figures must be improved.

Good point are discussion and conclusions , bringing many interesting findings.

Round 2

Reviewer 1 Report

The authors have improved their paper however there are minor comments which they must focus on.

1: Abstract must be concise, It should not be longer than 200-300 Words. 

2: Add related work from the articles about RBCs other than China as well. Do not just cite the article after one word. In response to my previous comment, your referred paragraph does not contain the related work. 

3: Conclusion should be the last section of your article. 

Reviewer 2 Report

The authors misunderstood the reviewers' comments, instead of making the paper more readable and succint, they added  a lot of text that makes the paper even more difficult to digest. 

The only added value of this paper is data analysis, methods are old and not innovative (rather old methodology), results are quite obvious with some exceptions. The paper is very long for such a limited contribution.

The paper is worth publishing but it must undergo a professional edition , substantial shortening and rewritten by a person who really knows English.

Round 3

Reviewer 2 Report

The paper has been improved according to reviewers' comments. Extensive rewriting lead to much more clear presentation and better presentation.